# Adaptive Backstepping Control of Quadrotor UAVs with Output Constraints and Input Saturation

**Jianming Li** , **Lili Wan \*** , **Jing Li and Kai Hou**

School of Electrical and Electronic Engineering, Wuhan Polytechnic University, Wuhan 430023, China; ljm1371979836@outlook.com (J.L.); hkcjj000999@outlook.com (K.H.)
* Correspondence: wanlili@whpu.edu.cn; Tel.: +86-18086061535

**Abstract:** The control performance of quadrotor unmanned aerial vehicles (UAVs) in complex environments can be affected by external disturbances and other factors. In this paper, an adaptive neural network backstepping controller based on the barrier Lyapunov function (BLF) is designed for a quadrotor UAV with internal uncertainties, input–output constraints and external disturbances. Radial basis function neural networks are used to approximate the uncertainties in the dynamic model of the UAV, while the minimum parameter learning method is combined to accelerate the adjustment speed of neural network weights. A robust term is designed to balance the total system disturbance and improve the anti-interference performance. The BLF is used to handle the output constraint so that the constrained parameters cannot break the predefined constraints. An auxiliary system is introduced to solve input saturation and avoid the dependence of tracking error on the input amplitude in the method of approximating input saturation using the smoothing function. The stability of the control system is demonstrated by the Lyapunov method. The simulation results show that the proposed method has high tracking accuracy compared with the backstepping dynamic surface control method, and the input and output are in the predefined range.

**Keywords:** quadrotor unmanned aerial vehicle; backstepping controller; output constraints; input saturation; barrier Lyapunov function

## 1. Introduction

The quadrotor unmanned aerial vehicle (UAV) is an unmanned flying machine composed of four propellers, which is attracting attention from various industries because of its small size, low cost, rapid take-off, vertical take-off, simple structure and landing and aerial hovering [1,2]. However, due to the inherent nonlinearity of the UAV system and the complexity of the operating environment, the UAV is inevitably affected by negative factors such as internal uncertainties, external disturbances and input–output constraints. Highly maneuverable flight and external disturbances make it easy for the UAV system inputs and outputs to reach the boundaries of the constraints. Therefore, it is a challenging task to ensure the trajectory tracking performance and stability of the UAV control system in a complex environment.

The UAV system is a typical nonlinear system. Currently, several methods have been used to design nonlinear system controllers, such as fuzzy control [3–5], robust H∞ control [6–8], adaptive control [9,10], proportional integral differential (PID) control [11], sliding-mode control [12–16] and backstepping control [17–20]. In the backstepping control, the complex system is divided into several subsystems, then Lyapunov functions and virtual control are designed for each external subsystem, and finally, the design of the whole control law is completed in a recursive way. However, traditional backstepping techniques cannot be used to design UAV control systems independently. The main problems are as follows: (1) inadequate anti-disturbance performance and (2) designing a higher-order system requires repeated differentiation of some functions, resulting in an increase in computational effort and a "complexity explosion".

Dynamic surface control (DSC) solves the "complexity explosion" by introducing a first-order filter in each step of the virtual control design [21,22]. In addition, the UAV in practical applications is affected by internal uncertainties and external disturbances. Several methods have been used to solve external disturbances, such as disturbance observer [23–25], expansion state observer [26] and adaptive control [27,28]. Among these anti-disturbance methods, adaptive control can effectively approximate and compensate unknown bounded disturbances and it is widely used in nonlinear systems. In [27], an adaptive command filtered backstepping sliding-mode control scheme was proposed for finite-time tracking control of quadrotor UAV systems. An adaptive control strategy was applied to estimate the upper bounds of the model uncertainties and external disturbances. Radial basis function (RBF) neural networks, which have good ability to approximate unknown functions, are widely used in nonlinear robust control system design [29]. In [30], an adaptive DSC based on quaternion was proposed to solve the problems of complexity explosion, internal uncertainties and external disturbances.

The above studies only considered the existence of external disturbances and internal uncertainties in the UAV system, but they did not take the effects caused by input–output constraints into consideration. The control input of the UAV may exceed the maximum generated by the actuators. Under the influence of external disturbances, such as high winds, the attitude of the UAV may sway too much or the fuselage can even flip. If the effects of attitude constraints and input saturation are ignored, the control performance of the UAV will be degraded.

Attitude constraints and input saturation are also essential factors affecting UAV control performance. Paper [31] used low-pass filters for the gain processing of nonlinear systems with input saturation. In [32], for multi-input multi-output stochastic nonlinear systems with full-state constraints and input saturation, a continuous differentiable asymmetric saturation model was proposed using a Gaussian error function representation. However, using smoothing functions to approximate input saturation in the above studies will result in the problem that tracking errors being affected by the amplitude of the reference signal. In [33], to deal with the UAV input saturation, the constrained command filter was introduced, and the auxiliary dynamic was designed to eliminate the effect of input saturation. Paper [34] investigated adaptive tracking control for stochastic nonlinear systems by introducing an auxiliary system for saturation compensation. This is a novel method of handling input saturation that explicitly constrains the tracking error. In addition, to prevent the UAV from being affected by external disturbances that cause flip-flopping, attitude constraints should be considered along with input saturation. In [35], the barrier Lyapunov function (BLF) was used to handle the full-state constraints. From the study, it can be found that BLF is an effective tool to deal with state constraints [36]. However, references [35,36] only considered output constraints and ignored the influence of input saturation. In summary, few studies considered input–output constraints for quadrotor UAVs with internal uncertainties and external unknown bounded disturbances. An in-depth study is necessary.

In this paper, we study the quadrotor UAV position–attitude trajectory tracking problem with external disturbances, internal uncertainties and input–output constraints. The main contributions of this paper are: (1) An adaptive neural network backstepping controller (ANNBC) in conjunction with DSC and an RBF neural network, and a robust term is proposed to deal with the position–attitude trajectory tracking. (2) A novel auxiliary system is introduced to solve the input saturation problem. This method can avoid the dependence of tracking error on the input amplitude in the method of approximating input saturation by using the smoothing function which is used in most of the literature. (3) A tandem control scheme is used to design a unified position–attitude controller, and BLF is combined to deal with output constraints.

The remainder of the article is organized as follows. Section 2 gives the quadrotor UAV dynamic equations and preliminaries. Section 3 proposes an ANNBC combined BLF, adaptive neural network, robust term and anti-saturation auxiliary system, and stability

analysis is given. Section 4 verifies the effectiveness of the proposed method through comparative simulation experiments. The conclusions and recommendations for future work are provided in Section 5.

## 2. Problem Formulation and Preliminaries

A quadrotor UAV is a flying machine with six degrees of freedom, consisting of four rotors arranged in a cross structure. As shown in Figure 1, the coordinate system of a quadrotor UAV is divided into an airframe coordinate system $E_B = \{O_B, X_B, Y_B, Z_B\}$ and an Earth coordinate system $E_E = \{O_E, X_E, Y_E, Z_E\}$. $[x, y, z, \phi, \theta, \psi]$ are the six states of the UAV, where $[x, y, z]$ are the position coordinates of the UAV relative to Earth, $[\phi, \theta, \psi]$ are Euler angles of the UAV's attitude relative to the body. The Euler angles around the $x$, $y$ and $z$ axes are expressed as the transverse roll angle $\phi$, pitch angle $\theta$ and yaw angle $\psi$, respectively. $F_1, F_2, F_3$ and $F_4$ represent the four lift forces generated by the four rotors, respectively [37].

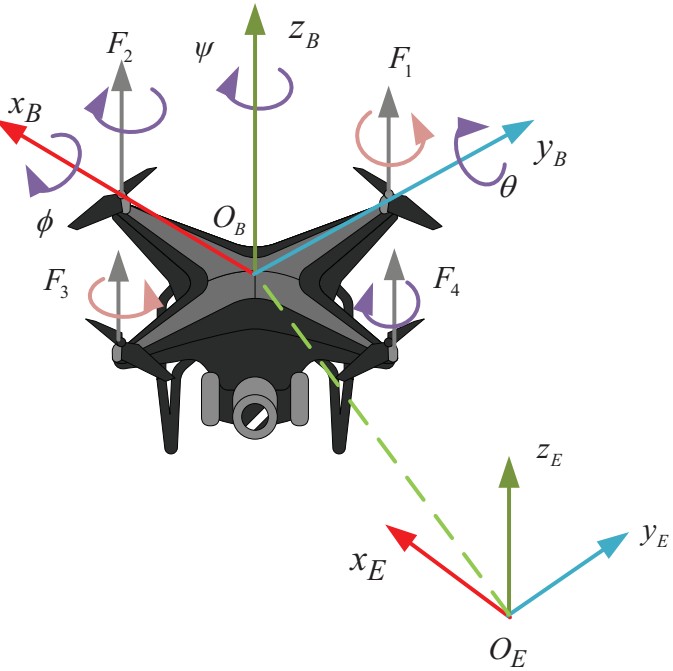

**Figure 1.** Coordinate systems of the quadrotor UAV.

The following assumptions are made for the quadrotor UAV dynamics model: (1) The quadrotor UAV is a standard cross structure and symmetrical, while the airframe is rigid. (2) The center of mass and center of gravity of the body structure of the UAV remain the same. (3) When the UAV is flying or aerial hovering, the angular velocity of the UAV's body coordinate system and the earth coordinate system are the same. Then, the dynamics model of the quadrotor UAV is represented as [38]

$$
\begin{cases}
\ddot{x} = \dfrac{1}{m_s}(\cos\phi\sin\theta\cos\psi + \sin\phi\sin\psi)U_1 - \dfrac{k_1\dot{x}}{m_s} \\[2mm]
\ddot{y} = \dfrac{1}{m_s}(\cos\phi\sin\theta\sin\psi - \sin\phi\cos\psi)U_1 - \dfrac{k_2\dot{y}}{m_s} \\[2mm]
\ddot{z} = \dfrac{1}{m_s}(\cos\phi\cos\theta)U_1 - g - \dfrac{k_3\dot{z}}{m_s} \\[2mm]
\ddot{\phi} = \dot{\theta}\dot{\psi}\dfrac{J_y - J_z}{J_x} + \dfrac{l}{J_x}U_2 - \dfrac{k_4 l}{J_x}\dot{\phi} \\[2mm]
\ddot{\theta} = \dot{\phi}\dot{\psi}\dfrac{J_z - J_x}{J_y} + \dfrac{l}{J_y}U_3 - \dfrac{k_5 l}{J_y}\dot{\theta} \\[2mm]
\ddot{\psi} = \dot{\phi}\dot{\theta}\dfrac{J_x - J_y}{J_z} + \dfrac{1}{J_z}U_4 - \dfrac{k_6}{J_z}\dot{\psi}
\end{cases}
\tag{1}
$$

where $m_s$ denotes the total mass of the quadrotor. $k_1, k_2, k_3, k_4, k_5$ and $k_6$ denote the drag coefficients of each channel, respectively. $J = diag\{J_x, J_y, J_z\}$ denote the axial rotational inertia of $x, y, z$ axes. $g$ denotes the acceleration of gravity. $l$ denotes the distance from the center of rotor to the center of the quadrotor. $(U_1, U_2, U_3, U_4)$ indicate the control inputs of the independent channels.

The UAV control system is divided into the position control subsystem and attitude control subsystem. The position and attitude control subsystem is divided into 6 control channels, and each of them is a standard quadratic nonlinear system. To facilitate the design of the controller, we simplify the dynamics model. The UAV has internal uncertainties. We define $f_i(X)$ to represent the unmodeled and uncertain part. In addition, the gyroscope torque is included in the total system perturbation. Considering the input–output constraints, the model uncertainties and external disturbances, a uniform nonlinear state expression for the 6 control channels of the UAV is derived as

$$\begin{cases} \dot{x}_{i1} = x_{i2} \\ \dot{x}_{i2} = b_i u_i + f_i(X) + w_i \end{cases}, i = x, y, z, \phi, \theta, \psi \tag{2}$$

where $X = [x_{x1}, x_{x2}, x_{y1}, x_{y2}, \ldots, x_{\psi 1}, x_{\psi 2}]^T = [x, \dot{x}, y, \dot{y}, z, \dot{z}, \phi, \dot{\phi}, \theta, \dot{\theta}, \psi, \dot{\psi}]^T$. The mass measurement and rotational inertia measurement errors of the UAV are uncertain, so the control gain $\bar{b}_i = [\frac{1}{(m_s+\Delta m_s)}, \frac{1}{(m_s+\Delta m_s)}, \frac{1}{(m_s+\Delta m_s)}, \frac{1}{(j_x+\Delta j_x)}, \frac{1}{(j_y+\Delta j_y)}, \frac{1}{(j_z+\Delta j_z)}]^T$ is uncertain. The measurement errors are divided into a deterministic part and an uncertain part. The control gain is rewritten as $\bar{b}_i = b_i + (\bar{b}_i - b_i)$. The deterministic part $b_i = [U_1/m_s, U_1/m_s, \cos\phi\cos\theta/m_s, 1/J_x, 1/J_y, 1/J_z]^T$ is involved in the UAV controller design, and the uncertain part is divided into a unknown function. $w_i$ denotes the bounded total perturbation of each channel, and the perturbation will be compensated by designing a robust term. $u_i$ indicates the control input for each channel.

The attitude control subsystem requires output constraints. When $i = \phi, \theta, \psi$, the attitude is constrained to the set $|x_{i1}| < k_{bi}$, where $k_{bi}$ is a constant. In addition, the control input obeys the following saturation properties:

$$u(v) = \text{sat}(v) = \begin{cases} u_M, & v \geq u_M \\ v, & u_m < v < u_M \\ u_m, & v \leq u_m \end{cases} \tag{3}$$

where $v$ denotes the unconstrained actual input. $u_M > 0$ and $u_m < 0$ denote the maximum and minimum of the control input, respectively.

**Assumption 1.** *For all $t > 0$, the desired tracking trajectory $x_{i1d}$ and its first and second order derivative are bounded, and they satisfies $\Omega_d = \{x_{i1d} \mid |x_{i1d}| \leq E_0, |\dot{x}_{i1d}| \leq E_1, |\ddot{x}_{i1d}| \leq E_2\}$, where $E_0, E_1, E_2$ are all positive constants.*

**Assumption 2.** *The approximation error $\varepsilon$ is bounded, i.e., $|\varepsilon| < b$, and $0 < b < 1$ is of arbitrary accuracy.*

**Assumption 3.** *The position and speed of the UAV are measurable.*

**Assumption 4.** *For all $t > 0$, both the uncertain internal part of the model and the external disturbances are bounded and there exists an unknown parameter d such that $\|w_i\| \leq d$.*

**Lemma 1** ([39])**.** *For any constant $k_{ai}$, let the open set $N = R \times Z \subset R^{l+1}$ and the interval $Z = \{e_{i1} \in R : -k_{ai} < e_{i1} < k_{ai}\} \subset R, k_{ai} \in R^+$, and consider the following system*

$$\dot{\zeta} = \vartheta(t, \zeta) \tag{4}$$

*where $\zeta = [\omega, e_{i1}]^T \in N$ and $\vartheta = R^+ \times N \to R^{l+1}$ are segmentally continuous at the time t and locally Lipschitz in the system $\zeta$, and consistent with t and $R^+ \times N$. Assuming that there exist functions $U : R^l \to R^+$ and $V_{i1} : Z \to R^+$, continuously differentiable and positive definite in their respective domains of definition. We obtain $V_{i1}(e_{i1}) \to \infty$ when $e_{i1} \to -k_{ai}$, or $e_{i1} \to k_{ai}$ and $\beta_1(\| \omega \|) \leq U(\omega) \leq \beta_2(\| \omega \|)$, where $\beta_1$ and $\beta_2$ are $K_\infty$ class functions. Assume that $V_{i1}(\zeta) = V_{i1}(x) + U(\omega, t)$, and $e_{i1}(0) \in Z$. If the inequality holds,*

$$\dot{V} = \frac{\partial V}{\partial \zeta} \vartheta \leq -DV + H \tag{5}$$

*where D and H are positive constants. Then, $e_{i1}$ remains in the set $e_{i1}(t) \in Z$, where $t \in [0, +\infty)$.*

**Lemma 2** ([34]). *For any positive constant $k_{ai}$ and any variable $e_{i1}$, the equation $ln \frac{k_{ai}^2}{k_{ai}^2 - e_{i1}^2} \leq \frac{e_{i1}^2}{k_{ai}^2 - e_{i1}^2}$ holds when the condition $|e_{i1}| < k_{ai}$ is satisfied.*

**Lemma 3** ([40]). *For an initially bounded system, if there exists a $C^1$ continuous and positive definite Lyapunov function $V(x)$ satisfying $\pi_1(x) \leq V(x) \leq \pi_2(x)$ such that $\dot{V}(x) \leq -r_1 V(x) + r_2$, where $\pi_1(x), \pi_2(x) : R^n \to R$ are $K_\infty$ like functions and $r_1, r_2$ are positive constants, then the solution $x(t)$ of the system is consistently bounded.*

**Lemma 4** ([41]). *The following inequality holds for any $\varsigma$ and $\hbar \in R$ that $0 \leq |\hbar| - \hbar \tanh\left(\frac{\hbar}{\varsigma}\right) \leq k_p \varsigma$, where is a constant that satisfies the condition $k_p = e^{-(k_p+1)}$, i.e., $k_p = 0.2785$.*

**Lemma 5** ([42]). *RBF neural networks have the ability to approximate unknown functions. $F(X)$ can be described by RBF neural networks as*

$$F(X) = \xi^{*T} h(X) + \delta \tag{6}$$

$$h_i(X) = exp\left\{ -\frac{(X - b_j)^T (X - b_j)}{c_j^2} \right\}, j = 1, 2, \ldots, L \tag{7}$$

*where $F(X)$ is the unknown continuous function of the tarobtain approximation over a compact set $X \in A_x \subset R^p$. $h(X) = [h_1, \ldots, h_L]^T$ is the basis function vector. $L > 1$ is the number of neural network nodes. $\delta$ is the minimum approximation error caused by RBF neural network. X is the input of the neural network. $b_j$ is the center value. $c_j$ is the width of the Gaussian function. $\xi^* \in R^L$ is the ideal weight, which defined as follows*

$$\xi^* = arg \min_{\hat{\xi} \in R^L} \left\{ \sup_{x \in A_X} \left| f(X) - \hat{\xi}^T h(X) \right| \right\} \tag{8}$$

*where $\hat{\xi} = \left[\hat{\xi}_1, \hat{\xi}_2, \ldots, \hat{\xi}_L\right]^T$ is the estimated weight vector.*

## 3. Adaptive Backstepping Controller Design

In this section, a BLF-based ANNBC with DSC, adaptive neural network and anti-saturation strategy is designed for the UAV control system with internal uncertainties, external disturbances and input–output constraints. A stability proof is given.

Due to the presence of four inputs and six outputs for UAVs, it is difficult to perform a unified controller design directly. In addition, there is a strong coupling relationship between each control output of the UAV. As shown in Figure 2, a tandem control scheme is used to design the UAV controller. The controller adopts double closed-loop control. The outer loop is the position controller, and the inner loop is the attitude controller. The desired trajectory is $(x_d, y_d, z_d)$ and $\psi_d$, and the desired attitude angles $[\phi_d, \theta_d]$ are computed by the position controller. The attitude tracking is achieved through the inner loop. RBF

neural networks are used to compensate system uncertainties and they are trained online according to tracking errors $(e_x, e_y, e_z, e_\phi, e_\theta, e_\psi)$. $\left(\hat{f}_x, \hat{f}_y, \hat{f}_z, \hat{f}_\phi, \hat{f}_\theta, \hat{f}_\psi\right)$ represent the outputs of the RBF neural networks.

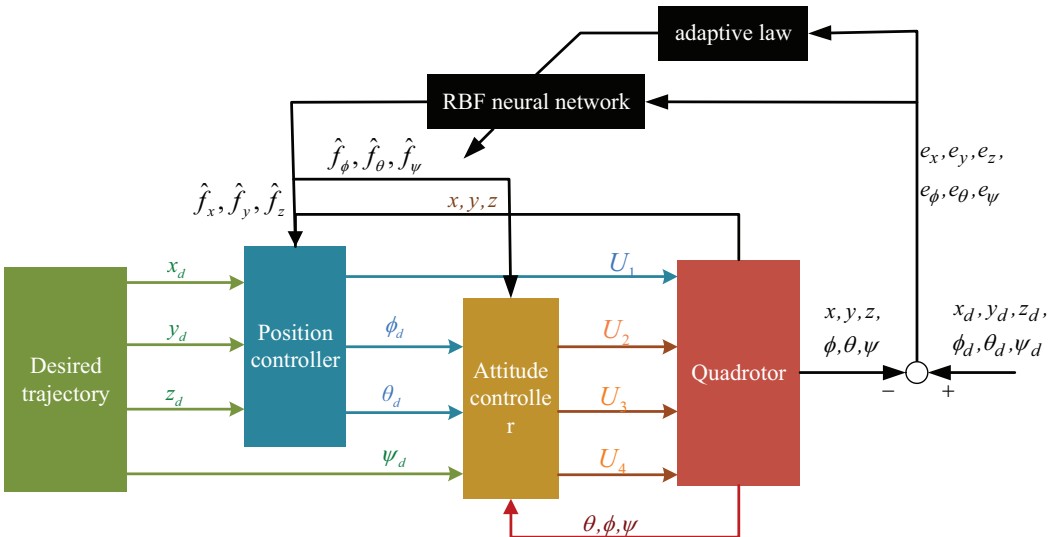

**Figure 2.** UAV tandem control scheme.

It can be seen from Figure 2 that $(\phi_d, \theta_d)$ are unknown. Two intermediate variables $u_x, u_y$ are introduced to calculate $(\phi_d, \theta_d)$. The intermediate variables $u_x, u_y$ are defined as

$$\begin{cases} u_x = \sin\psi\sin\phi + \cos\psi\sin\theta\cos\phi \\ u_y = \sin\psi\sin\theta\cos\phi - \cos\psi\sin\phi \end{cases} \tag{9}$$

According to Equation (9), we can obtain

$$\begin{cases} \phi_d = \arcsin\left(u_x\sin\psi - u_y\cos\psi\right) \\ \theta_d = \arcsin\dfrac{u_x\cos\psi + u_y\sin\psi}{\cos\phi_d} \end{cases} \tag{10}$$

where $u_x$ and $u_y$ are the control law of $x$ and $y$ channels, respectively. $\psi$ is the actual yaw angle of the UAV, which can be obtained by sensors.

**Remark 1.** *For constrained nonlinear systems, input saturation is approximated by using a smoothing function in [31,32]. The method will make the tracking error affected by the amplitude of the reference signal. Take a simple system $dx = (Px + u(v))dt$, for example, assuming that $P = 2$, $u_M = 2, u_m = -2, |x(0)| \le 2, y = x$, we can obtain $|y(t)| \le e^{-t}|x(0)| + (1 - e^{-t})u_M \le 4$ by integration. At this time, if the amplitude of the smoothing function $y_r(t) = 8$, then $|y(t) - y_r(t)| \ge 4$. It means that when the amplitude of the reference signal is larger, the tracking error will become larger. It has an impact on the tracking performance of the system.*

This paper applies a new anti-saturation strategy to the quadrotor UAV to solve the input saturation problem. Each channel of the UAV dynamics model is a second-order nonlinear system, so consider the following anti-saturation auxiliary system

$$\begin{cases} d\lambda_{i1} = (\lambda_{i2} - p_{i1}\lambda_{i1})dt \\ d\lambda_{i2} = (\Delta u - p_{i2}\lambda_{i2})dt \end{cases} \tag{11}$$

where $\lambda_{i1}, \lambda_{i2}$ is the design variable of the auxiliary system and $\lambda_{i1}(0) = 0, \lambda_{i2}(0) = 0$. $p_{i1}$ and $p_{i2}$ are positive constants to be designed. $\Delta u$ is the input saturation error defined as $\Delta u = u(v) - v$.

### 3.1. Controller Design

The position controller and attitude controller are designed separately, and the design process is similar. The control inputs of position–attitude control systems need to be designed with anti-saturation strategies.

Define the system tracking errors as

$$\begin{cases} e_{i1} = x_{i1} - \lambda_{i1} - x_{i1d} \\ e_{i2} = x_{i2} - \lambda_{i2} - s_i \end{cases} \tag{12}$$

where $x_{i1d}$ is the desired trajectory of the system state $x_{i1}$. $s_i$ is the introduced first-order filter virtual function. The first-order filter structure is

$$\begin{aligned} \tau_i \dot{s}_i + s_i &= \alpha_i \\ s_i(0) &= \alpha_i(0) \end{aligned} \tag{13}$$

where $\tau_i$ is the time constant of the filter. $\alpha_i$ is the virtual control.

Define the first-order filter function error as $T_i = s_i - \alpha_i$, and its derivative is

$$\dot{T}_i = \dot{s}_i - \dot{\alpha}_i = \frac{\alpha_i - s_i}{\tau_i} - \dot{\alpha}_i = -\frac{T_i}{\tau_i} + \varrho(x_{i1}, x_{i1d}, \dot{x}_{i1d}, \ddot{x}_{i1d}, \lambda_{i1}, \lambda_{i2}) \tag{14}$$

where $\varrho(x_{i1}, x_{i1d}, \dot{x}_{i1d}, \ddot{x}_{i1d}, \lambda_{i1}, \lambda_{i2})$ is a continuous function and assume that there is an unknown boundary $\varrho^*$ such that $\varrho \leq |\varrho^*|$.

Derivation of the tracking error $e_{i1}$ yields

$$\begin{aligned} \dot{e}_{i1} &= \dot{x}_{i1} - \dot{\lambda}_{i1} - \dot{x}_{i1d} = x_{i2} - \dot{\lambda}_{i1} - \dot{x}_{i1d} \\ &= \lambda_{i2} + s_i + e_{i2} - \lambda_{i2} + p_{i1}\lambda_{i1} - \dot{x}_{i1d} \\ &= e_{i2} + s_i + p_{i1}\lambda_{i1} - \dot{x}_{i1d} \end{aligned} \tag{15}$$

The states $(x, y, z)$ do not need to be constrained in the position controller, while the states $(\phi, \theta, \psi)$ need to be constrained. The Lyapunov function is designed as

$$V_{i1} = \begin{cases} \dfrac{1}{2}e_{i1}^2 + \dfrac{1}{2}T_i^2, & i = x, y, z \\ \dfrac{1}{2}\ln\dfrac{e_{i1}^2}{k_{ai}^2 - e_{i1}^2} + \dfrac{1}{2}T_i^2, & i = \phi, \theta, \psi \end{cases} \tag{16}$$

where $k_{ai}$ is the upper bound of the attitude tracking error, which satisfies $|e_{i1}| < k_{ai}$.

To facilitate the derivation of subsequent formulas, a function $a_i$ is defined for the state constrained.

$$a_i = \begin{cases} 1, & i = x, y, z \\ \dfrac{1}{k_{ai}^2 - e_{i1}^2}, & i = \phi, \theta, \psi \end{cases} \tag{17}$$

Taking the time derivative of (16). According to $s_i = T_i + \alpha_i$, we obtain

$$\dot{V}_{i1} = a_i e_{i1} \dot{e}_{i1} + T_i \dot{T}_i = a_i e_{i1}(e_{i2} + T_i + \alpha_i + p_{i1}\lambda_{i1} - \dot{x}_{i1d}) + T_i\left(-\frac{T_i}{\tau_i} + \varrho_i\right) \tag{18}$$

According to the fundamental theorem of Young's inequality, suppose there exists an arbitrary positive constant $\sigma_i$ that satisfies the following inequality:

$$|e_{i1}T_i| \leq e_{i1}^2 + \frac{1}{4}T_i^2 \tag{19}$$

$$|T_i\varrho_i| \leq \frac{1}{2\sigma_i}T_i^2\varrho_i^2 + \frac{\sigma_i}{2} \leq \frac{1}{2\sigma_i}T_i^2\varrho_i^{*2} + \frac{\sigma_i}{2} \tag{20}$$

Substituting (19) and (20) into (18) yields

$$\dot{V}_{i1} \leq a_i e_{i1}(e_{i2} + \alpha_i + p_{i1}\lambda_{i1} - \dot{x}_{i1d}) + a_i\left(e_{i1}^2 + \frac{1}{4}T_i^2\right) - \frac{T_i^2}{\tau_i} + \frac{1}{2\sigma_i}T_i^2\varrho_i^{*2} + \frac{\sigma_i}{2}$$
$$\leq a_i e_{i1}(e_{i2} + \alpha_i + p_{i1}\lambda_{i1} - \dot{x}_{i1d}) - T_i^2\left(\frac{1}{\tau_i} - \frac{1}{4} - \frac{1}{2\sigma_i}\varrho_i^{*2}\right) + \frac{\sigma_i}{2} + a_i e_{i1}^2 \tag{21}$$

According to Equation (21), the virtual control is designed as follows:

$$\alpha_i = -k_{i1}e_{i1} + \dot{x}_{i1d} - p_{i1}\lambda_{i1} \tag{22}$$

where $k_{i1}$ is the parameter to be designed and satisfies $k_{i1} > 0$

Substituting (22) into (21) results in

$$\dot{V}_{i1} \leq -a_i(k_{i1}-1)e_{i1}{}^2 + a_i e_{i1}e_{i2} - T_i^2\left(\frac{1}{\tau_i} - \frac{a_i}{4} - \frac{1}{2\sigma_i}\varrho_i^{*2}\right) + \frac{\sigma_i}{2} \tag{23}$$

Taking the time derivative of $e_{i2}$ yields

$$\dot{e}_{i2} = \dot{x}_{i2} - \dot{\lambda}_{i2} - \dot{s}_i = b_i u_i + f_i(X) + w_i - \Delta u_i + p_{i2}\lambda_{i2} - \dot{s}_i$$
$$= b_i u_i + \xi_i^{*T}h_i + w_i - \Delta u_i + p_{i2}\lambda_{i2} - \dot{s}_i \tag{24}$$

The RBF neural network is used to approximate the unknown function $f_i(X)$.

$$f_i(X) = \xi_i^{*T}h_i(X) + \delta_i \tag{25}$$

where $\xi_i^*$ is the vector of ideal weights. $h_i(X)$ is the Gaussian basis function. $\delta_i$ is the minimum approximation error of the RBF neural network which used to approximate the unmodeled and uncertain part $f_i(X)$. To balance the effects of approximation errors and external disturbances, an adaptive robust term is defined to compensate for the perturbations, and it can also compensate for the approximation error of RBF neural network. The external disturbances are time-dependent and the state of each control channel is bounded, so the total system perturbation $w_i$ is also bounded. Suppose there exists an unknown constant $w_i^*$ as the boundary of the perturbation, $w_i \leq |w_i^*|$ is satisfied.

The neural network minimum parameter learning method is used to improve the speed of adjusting parameters; let $\Theta_i^* = \| \xi_i^* \|^2$. The Lyapunov function is designed as

$$V_{i2} = V_{i1} + \frac{1}{2}e_{i2}^2 + \frac{1}{2\beta_i}\tilde{\Theta}_i^2 + \frac{1}{2\rho_i}\tilde{w}_i^2 \tag{26}$$

where $\beta_i, \rho_i$ are the two positive constants to be designed. $\hat{\Theta}_i$ is the estimate of the neural network weights, and $\tilde{\Theta}_i = \Theta_i^* - \hat{\Theta}_i$ is the estimation error between the desired weights and the estimated weights. $\hat{w}_i$ is the estimate of the total perturbation of the system, and $\tilde{w}_i = w_i^* - \hat{w}_i$ is the estimation error between the actual perturbations and the estimated perturbations. According to Equation (26), we obtain

$$\dot{V}_{i2} = \dot{V}_{i1} + e_{i2}\left(b_i u_i + \xi_i^{*T}h_i + w_i - \Delta u_i + p_{i2}\lambda_{i2} - \dot{s}_i\right) - \frac{1}{\beta_i}\tilde{\Theta}_i\dot{\hat{\Theta}}_i - \frac{1}{\rho_i}\tilde{w}_i\dot{\hat{w}}_i \tag{27}$$

According to the fundamental theorem of Young's inequality, suppose there exists an arbitrary positive constant $\sigma_i$ that satisfies the following inequality

$$e_{i2}\xi_i^{*T}h_i \leq \frac{1}{4\sigma_i}e_{i2}^2\Theta_i^*h_i^Th_i + \sigma_i \tag{28}$$

Substituting (28) into (27) yields

$$\dot{V}_{i2} \leq - a_i(k_{i1}-1)e_{i1}^2 - \left(\frac{1}{\tau_i} - \frac{a_i}{4} - \frac{\varrho_i^{*2}}{2\sigma_i}\right)T_i^2 + \frac{\sigma_i}{2} + e_{i2}(a_ie_{i1} + b_iu_i - \Delta u_i + p_{i2}\lambda_{i2} - \dot{s}_i)$$

$$+ |e_{i2}|w_i^* - e_{i2}w_i^*\tanh\left(\frac{e_{i2}}{\varsigma_i}\right) + e_{i2}\hat{w}_i\tanh\left(\frac{e_{i2}}{\varsigma_i}\right) + \frac{1}{4\sigma_i}e_{i2}^2\hat{\Theta}_ih_i^Th_i \tag{29}$$

$$+ \frac{1}{4\sigma_i}e_{i2}^2\tilde{\Theta}_ih_i^Th_i + \sigma_i + e_{i2}\tilde{w}_i\tanh\left(\frac{e_{i2}}{\varsigma_i}\right) - \frac{1}{\beta_i}\tilde{\Theta}_i\dot{\hat{\Theta}}_i - \frac{1}{\rho_i}\tilde{w}_i\dot{\hat{w}}_i$$

According to Lemma 4, we further obtain

$$\dot{V}_{i2} \leq - a_i(k_{i1}-1)e_{i1}^2 - \left(\frac{1}{\tau_i} - \frac{a_i}{4} - \frac{\varrho_i^{*2}}{2\sigma_i}\right)\chi_i^2$$

$$+ e_{i2}\left(e_{i1} + b_iu_i - \Delta u_i + p_{i2}\lambda_{i2} - \dot{s}_i + \frac{1}{4\sigma_i}e_{i2}\hat{\Theta}_ih_i^Th_i + \hat{w}_i\tanh\left(\frac{e_{i2}}{\varsigma_i}\right)\right)$$

$$+ \frac{1}{4\sigma_i}e_{i2}\hat{\Theta}_ih_i^Th_i + \hat{w}_i\tanh\left(\frac{e_{i2}}{\varsigma_i}\right) + \frac{\tilde{\Theta}_i}{\beta_i}\left(\frac{\gamma_i}{4\sigma_i}e_{i2}^2h_i^Th_i - \dot{\hat{\Theta}}_i\right) \tag{30}$$

$$+ \frac{\tilde{\Theta}_i}{\beta_i}\left(\frac{\gamma_i}{4\sigma_i}e_{i2}^2h_i^Th_i - \dot{\hat{\Theta}}_i\right) + \frac{\tilde{w}_i}{\rho_i}\left(\rho_ie_{i2}\tanh\left(\frac{e_{i2}}{\varsigma_i}\right) - \dot{\hat{w}}_i\right) + 0.2785\varsigma_iw_i^* + \frac{3\sigma_i}{2}$$

According to Lyapunov's stability theorem, the control law $u_i$, the adaptive laws $\dot{\hat{\Theta}}_i$ and $\dot{\hat{w}}_i$ of the parameter are designed according to Equation (30)

$$u_i = b_i^{-1}\left(k_{i2}e_{i2} - a_ie_{i1} + \dot{s}_i - \frac{1}{4\sigma_i}e_{i2}\hat{\Theta}_ih_i^Th_i - \hat{w}_i\tanh\left(\frac{e_{i2}}{\zeta_i}\right) - \Delta u_i + p_{i2}\lambda_{i2}\right) \tag{31}$$

$$\dot{\hat{\Theta}}_i = \frac{\beta_i}{4\sigma_i}e_{i2}^2h_i^Th_i - \gamma_i\hat{\Theta}_i \tag{32}$$

$$\dot{\hat{w}}_i = \rho_ie_{i2}\tanh\left(\frac{e_{i2}}{\zeta_i}\right) - v_i\hat{w}_i \tag{33}$$

where $k_{i2}$, $\gamma_i$ and $v_i$ are the parameters to be designed and satisfy $k_{i2} > 0$, $\gamma_i > 0$ and $v_i > 0$. Substituting Equations (31)–(33) into (30), we obtain

$$\dot{V}_{i2} \leq - a_i(k_{i1}-1)e_{i1}^2 - \left(\frac{1}{\tau_i} - \frac{a_i}{4} - \frac{\varrho_i^{*2}}{2\sigma_i}\right)T_i^2 - \frac{\gamma_i}{2\beta_i}\tilde{\Theta}_i^2 - \frac{v_i}{2\rho_i}\tilde{w}_i^2$$

$$- k_{i2}e_{i2}^2 + \frac{3\sigma_i}{2} + 0.2785\varsigma_iw_i^* + \frac{\gamma_i}{2\beta_i}\Theta_i^{*2} + \frac{v_i}{2\rho_i}w_i^{*2} \tag{34}$$

### 3.2. Stability Analysis

The control law (31) and the adaptive laws (32), (33) are selected based on Assumptions 1–4. The following properties hold if the system satisfies the state constraints $-k_{bi} < x_{i1}(0) < k_{bi}$ and input saturation.

**Theorem 1.** *All closed-loop signals are bounded.*

**Theorem 2.** *The attitude system state does not exceed the set constrained condition* $-k_{bi} < x_{i1}(t) < k_{bi}, \forall t > 0$.

**Theorem 3.** *The output tracking error can be made to converge to the domain of zero by selecting suitable parameters and the convergence accuracy is affected by the parameters.*

To facilitate the subsequent proof, first define

$$K_i = \frac{1}{\tau_i} - \frac{a_i}{4} - \frac{\varrho_i^{*2}}{2\sigma_i} \tag{35}$$

Substituting (35) into (34), then

$$\dot{V}_{i2} \leq -a_i(k_{i1} - 1)e_{i1}^2 - K_i T_i^2 - \frac{\gamma_i}{2\beta_i}\tilde{\Theta}_i^2 - \frac{v_i}{2\rho_i}\tilde{w}_i^2 - k_{i2}e_{i2}^2 + \frac{3\sigma_i}{2} + 0.2785\varsigma_i w_i^* \tag{36}$$

According to Lemma 2, it can be deduced that

$$-\ln\frac{k_{ai}^2}{k_{ai}^2 - e_{i1}^2} \geq -\frac{e_{i1}^2}{k_{ai}^2 - e_{i1}^2} \tag{37}$$

According to Equation (37), for the attitude control subsystem that requires output constrained, we can further obtain

$$\dot{V}_{i2} \leq -(k_{i1} - 1)\ln\frac{k_{bi}^2}{k_{bi}^2 - e_{1i}^2} - K_i T_i^2 - \frac{\gamma_i}{2\beta_i}\tilde{\Theta}_i^2 - \frac{v_i}{2\rho_i}\tilde{w}_i^2 - k_{i2}e_{i2}^2 + \frac{3\sigma_i}{2} + 0.2785\varsigma_i w_i^* \tag{38}$$

Define $D_i = min\{2(k_{i1} - 1), 2k_{i2}, 2K_i, \gamma_i, v_i\}$, $H_i = \frac{3\sigma_i}{2} + 0.2785\varsigma_i w_{i2}^*$. Based on (26), we can obtain

$$\dot{V}_{i2} \leq -D_i V_{i2} + H_i \tag{39}$$

**Proof of Theorem 1.** Multiplying both sides of (39) by $e^{tD_i}$ at the same time, the following inequality is obtained

$$e^{D_i t}\dot{V}_{i2} \leq (-D_i V_{i2} + H_i)e^{D_i t} \tag{40}$$

Perform the integration operation on Equation (40):

$$\begin{aligned}
\frac{d}{dt}\left(e^{D_i t}V_{i2}\right) &\leq H_i e^{D_i t} \\
e^{D_i t}V_{i2} - V_{i2}(0) &\leq \frac{H_{i2}}{D_i}\left(e^{D_i t} - 1\right) \\
0 \leq V_{i2} \leq V_{i2}(0)e^{-D_i t} + \frac{H_i}{D_i}\left(1 - e^{-D_i t}\right) &\leq V_{i2}(0) + \frac{H_i}{D_i}
\end{aligned} \tag{41}$$

If $V_{i2}(0) \leq \ell$, we can obtain $V_{i2}(0) \leq \ell + \frac{H_i}{D_i}$. Based on (39), it is known that $V_{i2}$ is bounded. It can be determined that $e_{i1}, e_{i2}, \alpha_i, s_i$ are bounded. Also, $u_i, x_{i1}$ are bounded. Then, all variables are in the closed-loop control system. □

**Proof of Theorem 2.** $-k_{bi} < x_{i1}(0) < k_{bi}$ can be obtained according to Lemma 1. Since $x_{i1}(t) = e_{i1}(t) + x_{i1d}(t)$, it can be deduced that

$$-k_{ai} + x_{i1d}(t) < x_{i1}(t) < k_{ai} + x_{i1d}(t), \forall t > 0 \tag{42}$$

Then we infer that $-k_{bi} < x_{i1}(t) < k_{bi}, \forall t > 0$. □

**Proof of Theorem 3.** Based on Equation (26), it can be obtained that

$$\frac{1}{2}\ln\frac{k_{ai}^2}{k_{ai}^2 - e_{i1}^2} \leq \left(V_{i2}(0) - \frac{H_i}{D_i}\right)e^{-D_i t} + \frac{H_i}{D_i} \tag{43}$$

Transforming both sides of Equation (43) into an exponent with $e$ as the base

$$\frac{k_{ai}^2}{k_{ai}^2 - e_{i1}^2} \leq e^{2\left[(V_{i2}(0) - H_i/D_i)e^{-D_i t} + H_i/D_i\right]} \tag{44}$$

Since $|e_{i1}| < k_{ai}$, it is known that $k_{ai}^2 - e_{i1}^2 > 0$, and it can be obtained that

$$|e_{i1}(t)| \leq k_{ai}(t)\sqrt{1 - e^{-2\left[(V_{i2}(0) - H_i/D_i)e^{-D_i t} + H_i/D_i\right]}} \tag{45}$$

When $t \to \infty, e_{i1}(t) \leq k_{bi}(t)\sqrt{1 - e^{-2H_i/D_i}}$, we obtain

$$|x_{i1}(t) - x_{i1d}(t)| \leq k_{ai}(t)\sqrt{1 - e^{-2H_i/D_i}} \tag{46}$$

According to the definitions of $H_i$ and $D_i$, the output tracking error converges to the domain of 0 when $t \to \infty$, and the convergence accuracy is affected by the parameters. □

## 4. Simulation Results

In this section, the proposed BLF-based ANNBC is applied to the quadrotor UAV to verify the anti-interference and tracking performance under constrained conditions. The superiority of the present method is verified through comparative simulation experiments with the backstepping dynamic surface control (BDSC) [43]. The controller design gives a uniform form of control law for each control channel of the UAV, and the parameters and constraints are different from each control channel. Considering the dynamics of equation (1), the parameters of the quadrotor UAV in [37] are listed in Table 1.

**Table 1.** Modeled parameters of the UAV.

| Parameter | Value | Units |
|:---:|:---:|:---:|
| $m_s$ | 0.468 | Kg |
| $l$ | 0.3 | m |
| $J_x, J_y, J_z$ | 0.0023 | $N \cdot m$ |
| $k_1, k_2, k_3$ | 0.01 | $N \cdot s/m$ |
| $k_4, k_5, k_6$ | 0.012 | $N \cdot s/rad$ |

The desired position–attitude trajectory of the UAV are as follows

$$[x_d(t), y_d(t), z_d(t)]^T = [t, 2\cos(0.5t), \sin(t)]^T \\ \psi_d(t) = 2\sin(t) \tag{47}$$

The initial conditions for the quadrotor UAV are given as

$$[x(t_0), y(t_0), z(t_0), \theta(t_0), \phi(t_0), \psi(t_0)]^T = [0, 0, 2, 0, 0, 0]^T \tag{48}$$

To make the comparison experiments more convincing, the parameters of the two control methods are identical. The relevant parameters of the control law (31), (22) and the adaptive law (32), (33) are given as $k_{x1} = 1, k_{x2} = 5, k_{y1} = 1, k_{y2} = 5, k_{z1} = 1, k_{z2} = 5$, $k_{\theta1} = k_{\theta2} = 10, k_{\phi1} = k_{\phi2} = 10, k_{\psi1} = k_{\psi2} = 10$. $\beta_i = 1, \gamma_i = 0.1, \rho_i = 1, v_i = 0.1, \varsigma_i = 0.01$, $\sigma_i = 1, \hat{\Theta}_i(t_0) = 0, \hat{w}_i(t_0) = 0$

Assume that the following uncertainties and external disturbances are [37]

$$\left[\Delta_m(t), \Delta_{Jx}(t), \Delta_{Jy}(t), \Delta_{Jz}(t)\right]^T = [0.2, 0.001 \times \sin(t), 0.001 \times \cos(t), 0.001(\sin(t) + 1)]^T, \\ [\Delta_{k1}, \Delta_{k2}, \Delta_{k3}, \Delta_{k4}, \Delta_{k5}, \Delta_{k6}] = 0.5 \times \text{rand}(1), \\ \left[w_x(t), w_y(t), w_z(t)\right]^T = [0.3 \times \sin(t), 0.3 \times \cos(t), 0.1 \times \sin(0.5t)]^T, \\ \left[w_\phi(t), w_\theta(t), w_\psi(t)\right]^T = [0.5 \times \cos(0.1t), 0.3 \times \sin(0.01t), 1 - 0.5 \times \sin(0.5t)]^T. \tag{49}$$

In this paper, we use RBF neural network to approximate the unknown part, and the selected structure is 2-5-1, which means two inputs, five hidden layer nodes, and one

output. The RBF neural network parameters are set as $c_x = c_y = c_z = c_\theta = c_\phi = c_\psi = \begin{bmatrix} -2, -1, 0, 1, 2 \\ -2, -1, 0, 1, 2 \end{bmatrix}$, $b_x = b_y = b_z = b_\phi = b_\theta = b_\psi = 3$. Set the attitude tracking error constraint to $k_{ai} = 0.51$. Using the anti-saturation strategy proposed in this paper, the maximum and minimum of the control inputs are set, respectively, as $U_{1M} = 9, U_{1m} = 4$, $U_{2M} = 5, U_{2m} = -10, U_{3M} = 2, U_{3m} = -5, U_{4m} = -30, U_{4M} = 30$.

Figures 3 and 4 represent the position-attitude tracking curves of the UAV using the ANNBC. Figures 5 and 6 represent the position–attitude tracking curves of the UAV using the BDSC. It can be found that in the presence of external disturbances and internal uncertainties, the ANNBC still has good control performance with small tracking errors. For the UAV using the BDSC in Figures 5 and 6, the outputs cannot track the desired trajectory well because there is no anti-disturbance design. Total disturbances and uncertainties severely impair the control performance and lead to large tracking errors. It is concluded that our proposed method has stronger anti-disturbance performance and better tracking performance.

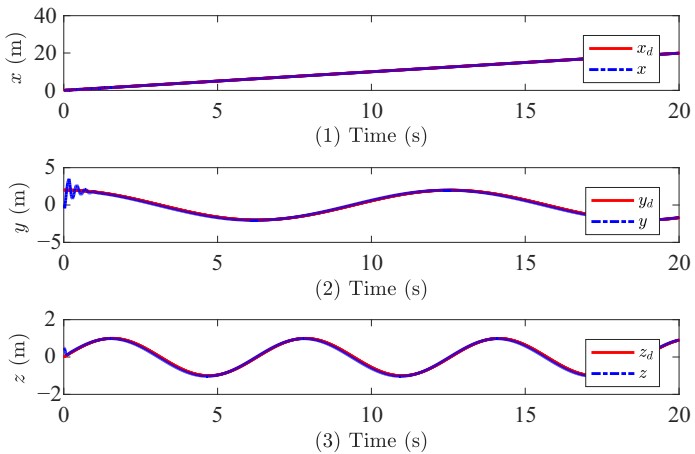

**Figure 3.** Response curves of position tracking of the ANNBC.

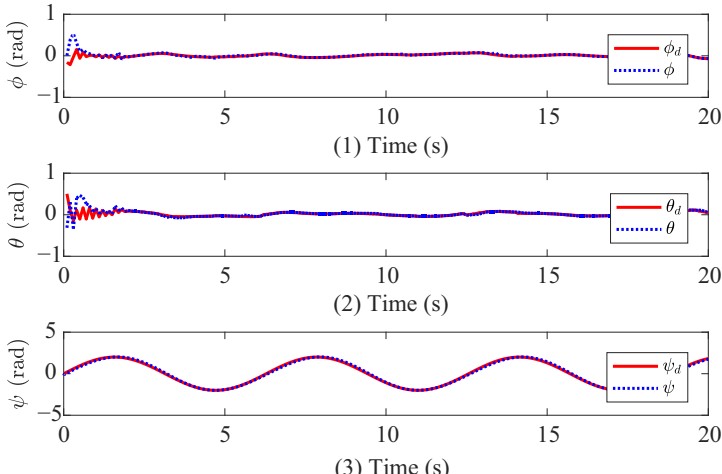

**Figure 4.** Response curves of attitude tracking of the ANNBC.

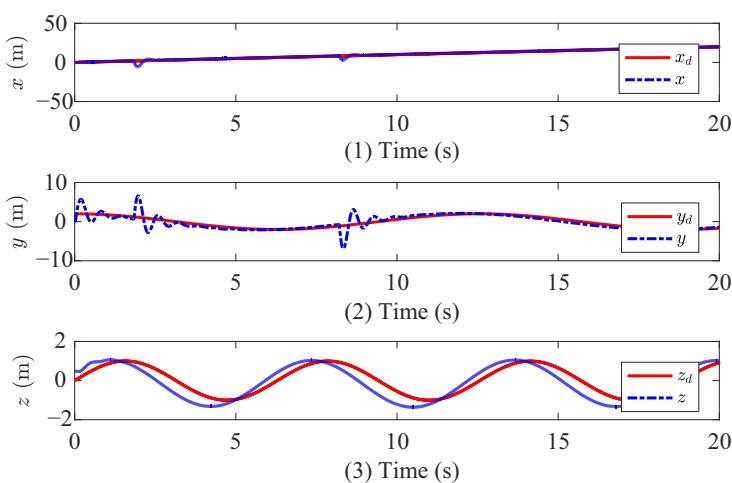

**Figure 5.** Response curves of positions tracking of the BDSC.

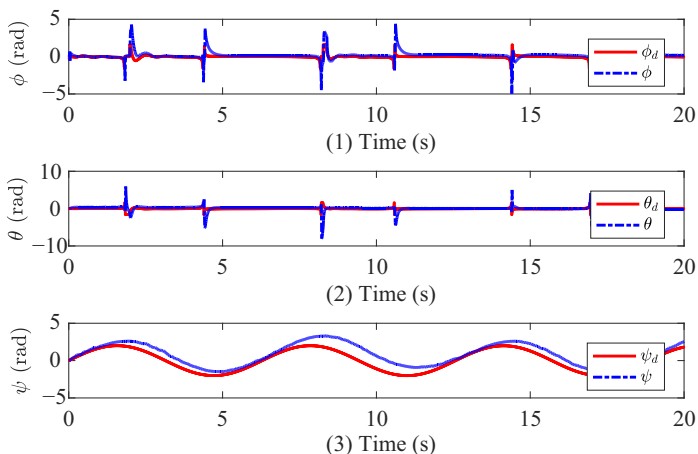

**Figure 6.** Response curves of attitude tracking of the BDSC.

The position–attitude tracking errors of the UAV control system using the ANNBC are shown in Figures 7 and 8, respectively. The position control subsystem has a large tracking error at the beginning. By introducing RBF neural networks and robust terms, the position-attitude tracking errors $[e_x, e_y, e_z, e_\phi, e_\theta, e_\psi]$ within the range of $[0.012, 0.057]$, $[0.021, 0.103]$, $[-0.072, 0.045]$, $[-0.013, 0.031]$, $[-0.015, 0.018]$ and $[-0.033, 0.002]$, respectively. The state of the attitude control channel needs to be constrained. It can be found that with ANNBC, the tracking error of the attitude control channel is always in the range of $[-0.51, 0.51]$, satisfying the set constraint. In the presence of external disturbances and internal uncertainties, our proposed ANNBC can achieve output constraints at all moments. While from Figures 5 and 6, since there is no anti-interference and output constraints design, small perturbation leads to a large tracking error. Output constraints are broken and the dynamic properties of the system are affected. It is concluded that the ANNBC has a very small steady-state error, which can well solve output constraints, model uncertainties and external disturbances , and meet the performance requirements of UAV for fast, accurate and anti-interference landing.

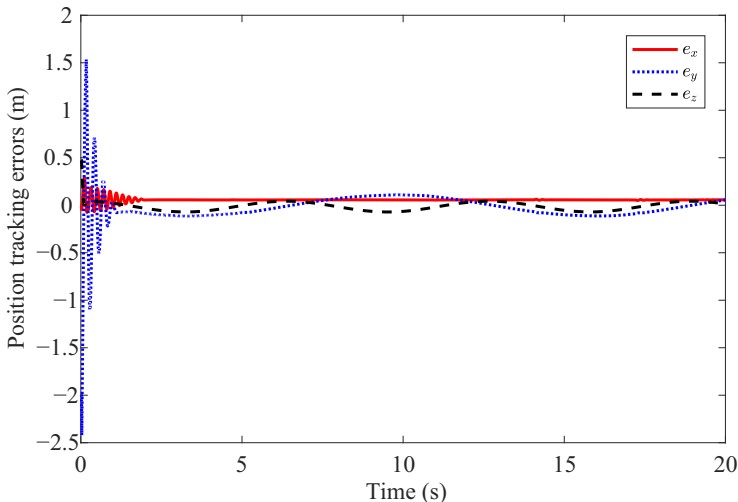

**Figure 7.** Curves of position tracking errors of the ANNBC.

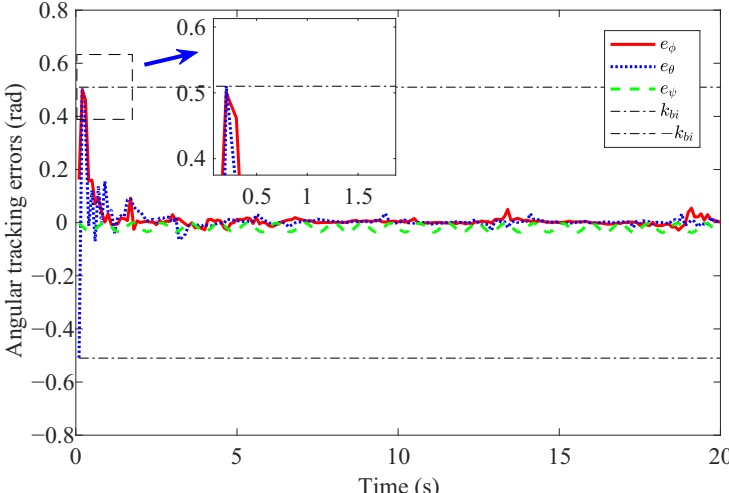

**Figure 8.** Curves of attitude tracking errors of the ANNBC.

In this paper, the maximum and minimum of the control inputs are set, respectively, as $U_{1M} = 9, U_{1m} = 4, U_{2M} = 5, U_{2m} = -10, U_{3M} = 2, U_{3m} = -5, U_{4m} = -30, U_{4M} = 30$. The curves of the control inputs without anti-saturation strategy $(v_1, v_2, v_3, v_4)$ and with the anti-saturation strategy $(U_1, U_2, U_3, U_4)$ are shown in Figure 9a–d. The UAV control system is a typical nonlinear system. As the control input is limited by physical factors, the control intput will show input saturation. To verify the effectiveness of the anti-saturation strategy, the control input without anti-saturation strategy and the control input with anti-saturation strategy are simulated for comparison. It can be seen that the control intput of each channel is always within the set constraints under the anti-saturation strategy. Control inputs without anti-saturation processing will be too large or too small at first. With the anti-saturation processing, the control intput is strictly limited to a constrained range and still has good tracking performance.

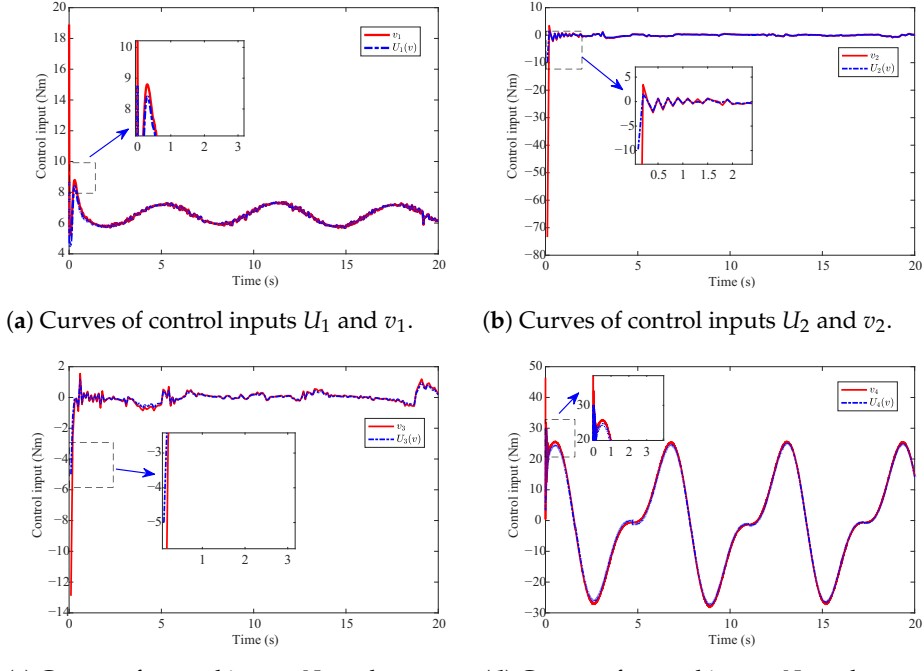

(**a**) Curves of control inputs $U_1$ and $v_1$.

(**b**) Curves of control inputs $U_2$ and $v_2$.

(**c**) Curves of control inputs $U_3$ and $v_3$.

(**d**) Curves of control inputs $U_4$ and $v_4$.

**Figure 9.** Curves of control inputs without anti-saturation strategy and with the anti-saturation strategy.

## 5. Conclusions

In this paper, we study the quadrotor UAV position–attitude trajectory tracking problem with external disturbances, internal uncertainties and input–output constraints. An ANNBC is proposed by combining BLF, DSC, an adaptive neural network and an anti-saturation auxiliary system. A tandem control scheme is adopted to design position and attitude controllers for the UAV. An adaptive neural network is used to approximate the uncertainty term of the UAV model. A robust term is designed to balance the total system disturbance. The BLF is used to cope with attitude constraints, while a new auxiliary system is introduced to solve the input saturation problem. Using the ANNBC, the UAV achieves high accuracy in tracking performance and input–output constraints. The stability of the system is guaranteed and all signals are bounded.

Due to limited conditions, the designed controllers in this paper are not simulated in a sufficient number of specific environments. In future research, we will consider validating the performance in more complex environments. In addition, we will further consider the UAV trajectory tracking problem with asymmetric time-varying output constraints, which is not considered in this paper.

**Author Contributions:** Methodology, J.L. (Jianming Li) and L.W.; software, J.L. (Jianming Li); validation, J.L. (Jianming Li), J.L. (Jing Li) and K.H.; writing—original draft preparation, J.L. (Jianming Li) ; writing—review and editing, J.L. (Jianming Li) and L.W.; supervision, L.W. All authors have read and agreed to the published version of the manuscript.

**Funding:** This research was funded by the Natural Science Foundation of Hubei Province grant number 2022CFB449.

**Institutional Review Board Statement:** Not applicable.

**Informed Consent Statement:** Not applicable.

**Data Availability Statement:** Not applicable.

**Conflicts of Interest:** The authors declare no conflict of interest.

## Abbreviations

The following abbreviations are used in this manuscript:

| | |
|---|---|
| $m_s$ | Mass |
| $g$ | Gravitational acceleration |
| $x_{i1} = [x, y, z, \phi, \theta, \psi]^T$ | Position and Euler angles |
| $x_{i2} = [\dot{x}, \dot{y}, \dot{z}, \dot{\phi}, \dot{\theta}, \dot{\psi}]^T$ | Linear velocity and angular velocity |
| $w_i = [w_x, w_y, w_z, w_\phi, w_\theta, w_\psi]^T$ | Total perturbation |
| $J = diag\{J_x, J_y, J_z\}$ | Axial rotational inertia |
| $E_B = \{O_B, X_B, Y_B, Z_B\}$ | Airframe coordinate system |
| $E_E = \{O_E, X_E, Y_E, Z_E\}$ | Earth coordinate system |

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
