# Peer review of "Adaptive Backstepping Control of Quadrotor UAVs with Output Constraints and Input Saturation"

_applsci, doi:10.3390/app13158710_

Round 1
Reviewer 1 Report
This paper studied the quadrotor UAV position-attitude trajectory tracking problem with external disturbances, internal uncertainties and input-output constraints. The study contains interesting findings and proposes a new method of introducing an auxiliary system to deal with UAV’s input saturation. This is a carefully done study and the findings are of considerable interest. Some comments are given to improve the quality of this paper.
In page 1, Abstract, the method for dealing with input saturation can be more specific. Maybe this part can be improved!
In page 2, Introduction, the description of reference (25) in the literature review can be more concise.
In page 2, line 76, consider using the past tense. Please pay particular attention to English grammar, spelling, and sentence structure so that the goals and results of the study are clear to the reader. Check the manuscript carefully.
In page 6, equation (11), the common form of the estimated weight should be .
In page 9, the form of the same variable must be correct and consistent, such as the estimate of the neural network weights in equation (29). The same problem also exists in equation (30) and (32). Please check the manuscript carefully.
In page 13, the captions of Figure 3, 4, 5 and 6 are inappropriate. A more common usage is “response curves of ......”
In page 15, response curves of attitude tracking errors in Figure 8 and response curves of attitude in Figure 4 do not match. Please check it.

There are a few expression issues in English that need to be corrected.
Reviewer 2 Report
This manuscript is clearly well written and organized. Here are my suggestions to improve the quality of this work.
1. Include some quantitative description of the achieved results such as the accuracy and stability.
2. The introduction is fine, some of the related work is missing and it should be included
https://doi.org/10.1016/j.ast.2019.05.032
doi.org/10.1155/2021/3997648
https://doi.org/10.3390/app12199538
3. Reduce the number of equations. Also recheck Eq.52
4. Reduce the number of figures. Combine multiple figs in single form
5. Mention the units in all the figures where necessary. Modify the font size for the text.
6. Compare your results with the previously reported in
https://doi.org/10.1016/j.ast.2019.05.032
Minor Grammatical corrections and spell check is required only.
Reviewer 3 Report
Please see in the attached file.

Please check typos.
Reviewer 4 Report
The here presented manuscript explores an adaptive neural network backstepping controller based on the barrier Lyapunov function (BLF) for improve UAV control performance. In my opinion this work and manuscript were conducted well and the proposed solution could be helpful to the scientific community and UAVs operators.
However, can be applied improvements and modifications based on the list below:
- Can the proposed solution be applied to other types of UAVs?
- In addition to the final simulation, I think that the real-time test in a specific environment is required.
- Furthermore I think that a discussion chapter should be included inside the manuscript for critical analysis on conducted experiment.
Thank You
Round 2
Reviewer 3 Report
Thank you very much for your responses. The paper is better to publish. Congratulations.